# Unravelling the Effects of the Healthy Primary School of the Future: For Whom and Where Is It Effective?

**DOI:** 10.3390/nu11092119

**Published:** 2019-09-05

**Authors:** Nina Bartelink, Patricia van Assema, Stef Kremers, Hans Savelberg, Dorus Gevers, Maria Jansen

**Affiliations:** 1Department of Health Promotion, Care and Public Health Research Institute (CAPHRI), Maastricht University, P.O. Box 616 6200 MD Maastricht, The Netherlands; 2Department of Health Promotion, School of Nutrition and Translational Research in Metabolism (NUTRIM), Maastricht University, P.O. Box 616, 6200 MD Maastricht, The Netherlands (S.K.) (D.G.); 3Academic Collaborative Centre for Public Health Limburg, Public Health Services, P.O. Box 33, 6400 AA Heerlen, The Netherlands; 4Department of Nutritional and Movement Sciences, Nutrition and Translational Research Institute Maastricht (NUTRIM), Maastricht University, P.O. Box 616, 6200 MD Maastricht, The Netherlands; 5Department of Health Services Research, Care and Public Health Research Institute (CAPHRI), Maastricht University, P.O. Box 616, 6200 MD Maastricht, The Netherlands

**Keywords:** intervention effects, microsystems, nutrition, parenting practices, physical activity, school health promotion

## Abstract

The ‘Healthy Primary School of the Future’ (HPSF) aims to integrate health and well-being within the whole school system. This study examined the two-year effects of HPSF on children’s dietary and physical activity (PA) behaviours at school and at home and investigated whether child characteristics or the home context moderated these effects. This study (*n* = 1676 children) has a quasi-experimental design with four intervention schools, i.e., two full HPSF (focus: nutrition and PA), two partial HPSF (focus: PA), and four control schools. Measurements consisted of accelerometry (Actigraph GT3X+) and questionnaires. Favourable effects on children’s dietary and PA behaviours at school were found in the full HPSF; in the partial HPSF, only on PA behaviours. Children in the full HPSF did not compensate at home for the improved health behaviours at school, while in the partial HPSF, the children became less active at home. In both the full and partial HPSF, less favourable effects at school were found for younger children. At home, less favourable effects were found for children with a lower socioeconomic status. Overall, the effect of the full HPSF on children’s dietary and PA behaviours was larger and more equally beneficial for all children than that of the partial HPSF.

## 1. Introduction

Dietary and physical activity (PA) habits are formed at a young age [1], whereby unhealthy habits can already lead to overweight and obesity [2]. Schools can play an important role in promoting healthy behaviours in children, since a significant proportion of a child’s day is spent there, and schools reach all children [3,4,5]. However, the school is one of the microsystems that interact to shape a child’s health and well-being [6,7]. This means that the impact of changes in the school may also interact with the child’s behaviour in other microsystems, e.g., the home context. This could lead to a transfer of improved health behaviours to the home context. However, compensatory behaviours might also occur: improvements in children’s health behaviours at school (extra PA or healthier dietary behaviours) may be compensated at home by, e.g., a decrease in PA or unhealthier dietary behaviours [8,9]. This school–home interaction might also cause different effects for children due to their home context. For example, a school-based intervention may be of greater benefit to those children with a high socioeconomic status (SES) background than to children with a low SES background, which leads to increased health inequities [10]. Moreover, parents can have different nutrition- and PA-related practices at home, e.g., parental behaviours and rules, which may also moderate the effects of school interventions on children’s health behaviours [6,7,11,12].

Additionally, not only might the home context moderate the effects of school health promotion efforts, child characteristics might also occur as an effect modifier [6]. Several reviews have stated that even though the intention of school health promotion efforts is to reach all children, specific subgroups of children often benefit more than others. The review by Stewart-Brown et al. [13] found gender-specific results in several studies: some school-based interventions showed larger improvements on PA and dietary behaviours in girls and others in boys. Age-specific effects were also found, with some interventions being more effective in older children and others in younger children [13]. The review by Cook-Cotton et al. indicated that overweight children may respond more slowly or less well to school-based interventions than other children [11].

The ‘Healthy Primary School of the Future’ (HPSF) is a Dutch initiative that aims to improve the health and well-being of all children in the school by sustainably integrating health and well-being within the whole school system [14,15]. The initiative is based on the principles of the Health Promoting School (HPS) framework, which includes a whole school approach, the participation of teachers, children and parents, and partnerships in the local community [16]. HPSF is being investigated in an overall study among four intervention and four control schools by a multi-disciplinary research group [14,15]. The overall study includes, among others, an extensive process evaluation and several effect evaluations. One of the effect evaluations found favourable intervention effects on children’s health behaviours in the two intervention schools that focused on both healthy nutrition and PA [17]. The two intervention schools that focused only on PA found no effects, also not on children’s PA behaviours. Since these results only presented overall effects, it is not known whether the effects occurred at school only or also at home. Nor is it known whether specific subgroups of children could be identified who benefitted more from HPSF in terms of PA and dietary behaviours [18].

The aim of the current study was to unravel the intervention effects of HPSF on children’s health behaviours. Two main research questions were formulated: (1) What is the effect of HPSF on children’s dietary and PA behaviours at school and at home? (2) Did child characteristics or the home context moderate the effects of HPSF on children’s dietary and PA behaviours?

## 2. Methods

### 2.1. Study Design

The current study is part of the overall study that investigates HPSF [15] and uses a longitudinal quasi-experimental design with four intervention schools and four control schools. All eight participating schools are situated in the Parkstad region in the southern part of the Netherlands. This region has a low average SES, and unhealthy behaviours and overweight are higher in prevalence compared with the rest of the Netherlands [19]. Inclusion criteria for the schools were being a member of the educational board ‘Movare’, since they were one of the initiators of HPSF, and a minimum of 140 children in the study years two till five, to be able to study the effects of HPSF with enough statistical power. Ethical approval (14-N-142) for the overall study was given by the Medical Ethics Committee Zuyderland located in Heerlen (Parkstad, the Netherlands). A detailed description of the overall study and the recruitment of the eight schools is reported in Willeboordse et al. [15].

### 2.2. The Healthy Primary School of the Future

Three cooperating organizations, i.e., the regional educational board ‘Movare’, the regional public health services and Maastricht University, developed the HPSF initiative [15]. In line with the HPS framework [16], the initiative intends to establish a co-creation movement in schools for the development and implementation of health-promoting changes in different aspects of the school system, i.e., the school’s physical and social environment, school’s health policy, education, and school routines. In addition to the HPS framework, the aim was to create some form of positive disruption in the school by initiating two changes top–down: (1) a free healthy lunch each day and (2) daily structured PA and cultural sessions after lunch, both implemented by external pedagogical employees provided by childcare organizations. These changes are adapted bottom–up and should lead to momentum for more bottom–up processes to create additional health-promoting changes in school [14]. Two of the four intervention schools decided to implement both changes and are referred to as the ‘full HPSF’. These schools also implemented additional health-promoting changes: they improved their health policy, e.g., policy regarding the consumption of water in school, they provided water bottles to all children, and have implemented an educational lunch once a week. The other two intervention schools only implemented the structured PA and cultural sessions and are referred to as the ‘partial HPSF’. These schools did not implement any additional health-promoting changes. Each school selected one teacher as school coordinator, who managed HPSF in their school and all four schools involved teachers and parents in the adoption decision and the process of adapting the changes into the school context. Implementation started in all the schools in November 2015. Overarching the four schools, the HPSF initiative was led by a project leader from Movare and an executive board with representatives of the three collaborating organizations, including the project leader. A project team was created with representatives of all partners involved: the four schools, Movare, regional Public Health Services, Maastricht University, the Limburg provincial authorities, childcare organizations, a caterer, and sports and leisure organizations.

### 2.3. Study Population

All children (aged 4 to 12) and their parents in the eight schools (*n* = 2326 at T0) were invited to participate in this study. This included children from study years one to eight. Recruitment was done via information brochures for parents. The research team visited the classrooms to inform children about this study and encouraged them to ask their parents to participate [15]. Parents had to sign an informed consent form to participate in all measurements for themselves and their child(ren). The group of children included in this study were: at baseline (T0,) children from study years one to seven; at T1, children from study years two to eight; and at T2, children from study years three to eight. Children who joined this study at T1 or T2 were included even though no baseline data were available. Children who switched to other schools between 2015 and 2017 were excluded.

### 2.4. Measures

Data were gathered annually during one week of measurements, conducted between September and November 2015 (T0, previous to the start of HPSF in November 2015), 2016 (T1) and 2017 (T2). Inter-rater variability was minimised by training researchers according to a strict protocol. The data collection and data processing were identical to the evaluation of the overall effects on children’s dietary and PA behaviours [17].

#### Potential Effect Modifiers

Child characteristics: Children’s study year and gender were collected via the database of the educational board Movare. Children’s weight status was assessed by measurements of their height and weight. BMI was determined and age- and gender-specific BMI cut-off points were used to define children’s weight status, i.e., non-overweight versus overweight (including obesity) [20].

Children’s socioeconomic background: A digital questionnaire for parents was used to obtain information about, among other things, children’s SES. This was calculated as the mean of standardised scores on maternal education level, paternal educational level, and household income (adjusted for household size) [21]. The mean scores were categorised into low, middle and high SES scores based on tertiles.

Patterns of health-promoting parenting practices at home: The digital questionnaire for parents was also used to assess parents’ nutrition-related practices (n = 9) and PA-related practices (n = 14), e.g., modelling behaviour and encouragement. The questions were based on previous work by Gevers et al. [22,23] and O’Connor et al. [24]. Each item described a practice by giving a statement, followed by some examples. Participants responded on a Likert scale from 1 (completely disagree) to 5 (completely agree). Two cluster analyses similar to Gevers et al. were conducted: one for the nutrition-related parenting practices and one for the PA-related parenting practices [22]. Detailed information of the results of each clustering is described in Appendix A. Clustering of the nutrition-related parenting practices showed similar clusters of parents compared to the study by Gevers et al. [22]. The names of the clusters were: Cluster 1 (*n* = 226; 36.9%) “High involvement and supportive”; Cluster 2 (*n* = 102; 16.7%) “Low covert control and non-rewarding”; Cluster 3 (*n* = 78; 12.7%) “Low involvement and indulgent”; and Cluster 4 (*n* = 206; 33.7%) “High covert control and rewarding”. Clustering of the PA-related parenting practices also resulted in four clusters. The names of the clusters were: Cluster 1 (*n* = 220; 35.0%) “High involvement and supportive”; Cluster 2 (*n* = 133; 21.2%) “Moderate involvement, indulgent of child’s sedentary activities”; Cluster 3 (*n* = 17; 2.7%) “Low involvement and indulgent”; and Cluster 4 (*n* = 258; 41.1%) “Moderate involvement, supportive of child’s sedentary activities”.

### 2.5. Outcomes

Children’s PA levels—accelerometry: At the beginning of the measurement week, all participating children from study years two to eight received an accelerometer for seven days (Actigraph GT3X+, 30 Hz, 10 s epoch). The monitor was attached to the hip with an elastic band and had to be worn all day except while sleeping or during activities in which water was involved (e.g., swimming, bathing and showering). The accelerometry data were processed using ActiLife version 6.13.3. Wear time validation was assessed using Choi’s classification criteria [25]. Minimal wear time was defined as 480 min per day between 06:00 and 23:00 [26]. The first day of measurement was excluded to prevent reactivity [27]. Measurements containing at least three weekdays (after excluding the first measurement day) and one weekend day were used in the analyses [28]. The activity levels, classified using Evenson’s cut-off points, were in counts per minute (CPM) [29]: sedentary behaviour (SB; ≤100 CPM), light PA (LPA; 101–2295 CPM), and moderate to vigorous PA (MVPA; ≥2296 CPM).

Children’s dietary behaviours—a parent-reported questionnaire: A digital questionnaire for parents was used to obtain information about their children’s dietary behaviours. All parents of participating children (study years one to eight) received the questionnaire. Twelve questions from the Local and National Youth Health Monitor were used to assess children’s dietary behaviours [30,31]. Parents were asked about the number of days during the past week their child had breakfast, ate warm vegetables, salads or raw vegetables, fruits, consumed water and sugar-sweetened beverages (soft, sports, and energy drinks), and ate the following four snack types: chocolate, salted snacks, cookies, and soft ice-creams. A total score for healthy dietary behaviours (in mean days/week) was calculated by the mean number of days children consumed breakfast, fruits, vegetables (warm and cold), and water. A total score for unhealthy dietary behaviours (in mean days/week) was calculated by the mean number of days children consumed sugar-sweetened beverages and the four different snack types.

Children’s dietary behaviours—two child-reported questionnaires: The first child questionnaire (for children of study years four to eight) was filled out by writing during class hours in the presence of at least one member of the research team. The questionnaire was used to assess, among other things, children’s school water consumption (0 (almost) never; —1 sometimes (1–3 days per week); —2 often (4–6 days per week); —3 every day) and their breakfast intake. Breakfast questions consisted of recall questions (7 items) with yes/no answer options regarding the consumption of healthy food items, i.e., bread, cereals, butter, cheese, fruits, milk/yoghurt, and water, during breakfast that day. The items bread and cereals were combined into the food type grains, and milk/yoghurt and cheese were combined into the food type dairy. To give an indication about how healthy the children’s breakfast was that day, the six different food types consumed were summed, and a dichotomous variable was created to study whether children consumed at least two of the food types during breakfast. The second questionnaire (for children of study years three to eight) was used to assess the children’s lunch intake. The questions (except for an extra question regarding vegetable consumption during lunch) and the processing of the data were similar to those for the breakfast intake.

### 2.6. Analyses

Data were analysed using IBM SPSS Statistics for Windows (version 23.0. Armonk, NY, USA: IBM Corp). Missing data, including missing data at baseline, were imputed using a multiple imputation method with fully conditional specification (FCS) and 10 iterations, generating 50 complete datasets. Linear mixed-model analyses (continuous outcomes) and generalised estimating equations (binary outcomes) were used (see Bartelink et al. [17]) for the overall outcomes and school- and home-specific outcomes. Since measurements were repeated within participants, we used a two-level model with measurements as the first level and participants as the second level. The fixed part of the model consisted of group (full HPSF, partial HPSF, control), time (T0, T1, T2) and the interaction terms of group with time. We were not able to include class as a level in the model, because commonly more than one division of a class existed, e.g., 4a or 4b, and children often did not have fixed class divisions for all years. To obtain children’s setting-specific PA behaviours, the overall accelerometry data were divided by filters on wear time during school hours (school-specific PA) and wear time outside of school hours (home-specific PA). Regarding children’s dietary behaviour, the two total scores (healthy and unhealthy dietary behaviours) were used as overall outcomes; children’s lunch intake and their school water consumption as school-specific outcomes; and children’s breakfast intake as home-specific outcomes. All analyses were adjusted for gender, study year at baseline, ethnicity, SES, and children’s BMI z-score at baseline. A two-sided p-value ≤0.05 was considered statistically significant. Standardised effect sizes (ES) were calculated, which were defined as estimated mean difference after two years divided by the square root of the residual variance at baseline (pooled over the intervention groups). Binary outcomes resulted in odds ratios (ORs). To investigate whether the intervention effects were similar for all children, the following potential effect modifiers were considered: gender (boys/girls), study year at baseline (lower (1–4)/higher (5–8) grades), baseline weight status (non-overweight/overweight), SES (low/middle/high), and the patterns of nutrition-related parenting practices (four clusters) and PA-related parenting practices (four clusters). To assess potential effect modification, the interaction term group*time*effect modifier, with all corresponding two-way interactions, was added to the model [17]. When this interaction term was significant (here, we used a significance level of 0.10 to deal with the fact that the power of a test for interaction is relatively low [32] and we did not want to miss any effect modification), the intervention effects were reported for all categories of the effect modifier separately.

## 3. Results

Of all the children (*n* = 2326) invited to participate in this study, 60.3% joined this study at baseline (n = 1403). Because of this study’s dynamic population, a total of 1974 children and their parents participated in this study within the two-year follow-up period (data collected at least at one time point). Due to the selection used for the current study, i.e., children were eligible for the current research only when they were in study years one to seven at baseline and did not switch schools, 1676 children were included in the analyses. See Appendix A for characteristics of the study sample.

### 3.1. Intervention Effects

#### 3.1.1. Intervention Effects at School

Both the full and partial HPSF resulted in significant favourable intervention effects on children’s PA behaviours at school (Table 1). The time children spent in MVPA at school had increased significantly more in the full HPSF (ES = 0.34) and partial HPSF (ES = 0.29) compared to the children in the control schools. Favourable trends were found for both sedentary time (full HPSF: ES = −0.20; partial HPSF ES = −0.17) and light PA at school (full HPSF: ES = 0.11; partial HPSF ES = 0.09). Regarding children’s dietary behaviours at school, several favourable significant intervention effects were found in the full HPSF compared to the control schools: more children increased their water consumption at school (ES = 1.14) and ate at least two food types during lunch (OR = 2.98). In the partial HPSF, no significant intervention effects were found on children’s dietary behaviours at school.

#### 3.1.2. Intervention Effects at Home

The results showed no statistically significant favourable or adverse (compensatory) intervention effects of the full HPSF on children’s PA and dietary behaviours at home (Table 1). In the partial HPSF, an adverse intervention effect was found at home for children’s PA behaviours: the time children spent in light PA at home had decreased significantly more compared to children of the control schools. No significant favourable or adverse intervention effect was found for their dietary behaviours at home.

### 3.2. Effect Modifiers

#### 3.2.1. Moderators of Overall Intervention Effects

Fewer moderators of intervention effects on children’s dietary and PA behaviours were found in the full HPSF compared to the partial HPSF (1 vs. 4 significant moderators; Table 2). In the full HPSF, no moderators of children’s dietary behaviours were found and one moderator of effects on children’s PA behaviours was found. Gender moderated the effect on MVPA, as boys had increased their time in MVPA significantly more compared to boys in the control schools (ES = 0.34); this effect was not found in girls (ES = 0.02). In the partial HPSF, study year and SES were found to be significant moderators. Study year moderated the intervention effects on the time children spent sedentary and in light PA, with a favourable trend found in older children and an adverse trend in younger children. SES moderated the effects on the time spent in MVPA and unhealthy dietary behaviours. Each SES tertile showed a different trend, with a lack of consistency. In both the full and partial HPSF, no moderation was found by children’s weight status and parenting practices.

#### 3.2.2. Moderators of Intervention Effects at School

Fewer moderators of intervention effects at school were found in the full HPSF compared to the partial HPSF (2 vs. 4 significant moderators; Table 3). In both the full and partial HPSF, intervention effects at school were moderated by study year. In the full HPSF, study year moderated the effects on sedentary time and light PA. In the partial HPSF, study year moderated the effect on sedentary time, light PA and school water consumption. Consistently throughout the subgroup analyses, it was found that in older children, the effects were more favourable than in younger children. No other moderations were found in the full HPSF. In the partial HPSF, school water consumption was significantly moderated by gender. An adverse significant intervention effect was found for girls (ES =−0.36), and a trend in a favourable direction was found for boys (ES = 0.26).

#### 3.2.3. Moderators of Intervention Effects at Home

Fewer moderators of intervention effects at home were found in the full HPSF than in the partial HPSF (3 vs. 5 significant moderators; Table 4). In both the full and partial HPSF, SES moderated the intervention effects on children’s PA behaviours at home. In general, adverse effects or trends at home were found for the children in the lowest SES tertile and favourable effects or trends for the children in the highest SES tertile. In the full HPSF, a significant favourable effect on the time spent sedentary (ES = −0.34) was found for the children in the highest SES tertile. In the partial HPSF, a significant adverse effect for sedentary time (ES = 0.46) and light PA (−0.52) was found for the children in the lowest SES tertile. Appendix A presents all pairwise comparisons. The comparison between the lowest and highest SES tertile consistently shows a different effect at home compared to the effects at school. The results showed consistent differences in effect between the lowest and highest SES tertiles regarding PA behaviours at home (B > 1 in 5 out of 6 comparisons, Appendix A), whereas no differences in effect were found at school (B < 1 in all six comparisons, Appendix A). As an exception, this pattern was not found for children’s breakfast intake. No other significant moderations were found in the full HPSF for children’s PA and dietary behaviours at home. In the partial HPSF, the intervention effects at home on sedentary time, light PA, and on the consumption of at least two food types during breakfast were also moderated by weight status. Favourable trends were found at home in overweight children and adverse trends or effects were found in non-overweight children.

## 4. Discussion

The current study aimed to unravel the effects of HPSF on children’s dietary and PA behaviours. It investigated the intervention effects of HPSF at school and at home. The results showed that the time children spent in MVPA at school had increased in both the full and partial HPSF. However, children of the full HPSF did not compensate at home for the improved health behaviours at school, while in the partial HPSF, the results indicated that the children did compensate by becoming less active at home. Children’s dietary behaviours at school improved in the full HPSF, without compensating for these improvements at home.

The current study also investigated whether child characteristics or the home context moderated the intervention effects of HPSF. The results showed that the effects in the partial HPSF were influenced more often by moderators (in total, 13 significant moderators (18.6%)) than in the full HPSF (in total, six significant moderators (8.6%)), which indicates that the full HPSF had a more equal beneficial effect for all children. The findings indicated that the intervention effects of HPSF on children’s PA and dietary behaviours were mainly moderated by SES and study year. This is in contrast to a previous study on the effects of HPSF on children BMI z-scores, in which no moderators emerged (18). An explanation for the contrasting results may be that the effect on children’s BMI z-score is a result of the co-existence and interaction of the children’s nutrition and PA behaviours [33]: a moderating impact on one health behaviour may therefore not automatically lead to moderation of intervention effects on children’s BMI z-score.

The effects on children’s PA behaviours at school were moderated by study year in both the full and partial HPSF: older children benefitted consistently more from HPSF than younger children. This may indicate that the activities in school were more appropriate for the older children. This is in line with the results and conclusions of several studies that indicated that children of different ages have different needs regarding PA activities [6,34]. Previous research has for example shown that older children’s activity levels were more negatively affected by the number of peers present, while younger children were more negatively affected by the number of supervisors [35]. Therefore, it is recommended when implementing PA-related activities to ensure that either they are appropriate for all children or that age-specific PA activities are implemented.

The findings in this study showed that HPSF succeeded in creating equal effects on children’s PA and dietary behaviours at school, independent of the children’s backgrounds (SES, parenting practices). However, the findings also showed that the children’s socioeconomic background did influence the effects at home. The children with the lowest SES scores did not improve their PA behaviours at home; results even showed compensating behaviours of PA in these children at home. For the children from the highest SES group in the full HPSF, however, a transfer of the effects on PA was found from school to the home context. This suggests that the changes in the full HPSF schools have led to such an impact that these children also engaged in more PA after school. In contrast, these favourable effects at home did not occur in the children with the lowest SES scores. The compensation of PA at home in the lowest SES group has led to opposite effects at school and at home for these children: at school, their PA behaviours became more favourable; at home, they became less favourable. Since these opposite effects were especially found for the children with the lowest SES scores, it may contribute to an increased socioeconomic health equity gap. In addition, the partial HPSF also showed a moderating effect of SES on children’s overall dietary behaviours, with less favourable, and even adverse effects for the children with lower SES scores. Children in these schools, in contrast to children in the full HPSF, brought all foods and drinks that they consumed at school from home. This means that the dietary behaviours that are included in this outcome are actually dietary behaviours that have their origin in the home context. The moderating effect of SES on this outcome in the partial HPSF seems to indicate that not only children’s PA behaviours at home, but also their dietary behaviours at home are moderated by SES, with consistently less favourable effects for children with lower SES scores. It should be noted though that since all the schools are located in a low SES area, the SES tertiles used in this study are relative and not absolute scores. This means that the average SES score of children in this study sample is low compared to the average of the Netherlands [36]. Nevertheless, the differences in effect at home among children with lower and higher socioeconomic background demonstrate the interaction between two main microsystems of a child, i.e., school and home. This underlines that the school is an open system and interacts with other microsystems, such as home or the neighbourhood, to shape a child’s health and well-being [6,7]. According to a study by Gubbels et al., larger consistency across microsystems leads to more favourable effects on children’s health behaviours [37]. Therefore, it can be recommended for school health promotion efforts to include the home context in the HP changes, e.g., homework assignments that include parents, and/or to focus directly on health promotion in the home context [38]. In this way, the child’s environment enables healthier choices both at school and at home. This creates more consistency between the different microsystems of a child, particularly for children with lower socioeconomic backgrounds [39].

No moderating effect of parenting practices was found in the current study, which indicates that the effects of HPSF were not strengthened or weakened by parenting practices at home. This is in line with the results of a previous Dutch study conducted in secondary schools, which investigated whether family environmental factors affected changes in adolescent’s dietary behaviour who participated in a school health promotion program [40]. These findings indicate that it is not so much the parenting practices that explain the differences in home effects across SES groups, but that other aspects in the home context, such as the physical environment in the neighbourhood, may explain these differential intervention effects at home.

### Strengths and Limitations

The quasi-experimental design can be seen as a limitation of this study, since we were unable to (cluster-) randomise schools. To deal with this limitation, we controlled for BMI z-score at T0, gender, study year at T0, SES score, and ethnicity in all analyses. However, despite the lack of randomization, the design enabled us to test the effectiveness in terms of differences in children’s health behaviours between the three school groups over time, and we were able to enrol schools on the basis of motivation, which reflects the real-life situation of school health promotion. Another limitation is the multiple statistical testing in this study, which may increase the likelihood that the observed statistical differences have arisen by chance. Methodological strengths of this study include the objectively measured PA levels, all collected in the same season, and the matching of all measurements in the same week to prevent overburdening of the people in the school. Furthermore, since we had insight into the specific school hours of all included schools, we were able to separate children’s PA behaviours at school and at home. Another strength of this study is that child-reported data were collected regarding both their breakfast and lunch. This created the possibility to investigate the effects of HPSF on specific dietary behaviours of children at school and at home. It should be mentioned though that by categorizing the consumption of breakfast as a dietary behaviour at home, the assumption was made that children consumed their breakfast at home. Even though this is very common in the Netherlands, we do not have the data to confirm this assumption. Moreover, to investigate as much as possible children’s actual dietary behaviours during these meals, the answers of children without interference of parents were used. The specific effects of HPSF on both meals became more visible due to the use of comparable questions. In general, the assessment of children’s dietary behaviours had its limitations as only questionnaires were used, which are subjective measurements and may lead to socially desirable answers [41]. To assess children’s dietary behaviours more objectively, future research could include image collection methods. The parenting practices questionnaire, which also had these subjectivity limitations, was based on a validated questionnaire by Gevers et al. [23]. However, we were not able to include all practices described by Gevers et al., due to limitations in the length of the questionnaire as many other aspects were included in this questionnaire. The reduction in assessed practices was based on expert judgement; practices were only deleted when they were more or less similar to another practice. Due to this systematic approach, we were still able to conduct a cluster analysis, which led to the same four clusters [22]. The findings of this cluster analysis seem to indicate that the four patterns are also visible in another study sample, which is a next step in the validation of these patterns of parenting practices. Future research is needed to validate these findings and to investigate whether these clusters are also applicable in other study samples.

## 5. Conclusions

The effect of the full HPSF on children’s dietary and PA behaviours was not only larger, but also more equally beneficial for all children than that of the partial HPSF. In both the full and partial HPSF, less favourable effects at school were found for younger children. At home, less favourable effects were found for children with lower SES scores. It is recommended to include the home and neighbourhood context in health promotion efforts in order to create more consistency across the different microsystems of a child. This may particularly benefit children from lower socioeconomic backgrounds.

## Figures and Tables

**Table 1 nutrients-11-02119-t001:** Intervention effects of Healthy Primary School of the Future (HPSF) at school and at home.

	Full HPSF vs. Control			Partial HPSF vs. Control		
	B (95% C.I.)	*p*	ES	B (95% C.I.)	*p*	ES
**Overall PA and Dietary Behaviours ***						
Sedentary (% per day)	−1.29 (−2.39–−0.19)	**0.02**	−0.23	−0.17 (−1.25–0.90)	0.76	−0.03
Light PA (% per day)	0.94 (0.07–1.81)	**0.03**	0.22	−0.03 (−0.88–0.82)	0.95	−0.01
MVPA (% per day)	0.36 (−0.10–0.82)	0.12	0.15	0.19 (−0.26–0.64)	0.41	0.08
Healthy dietary behaviours (mean days/week)	0.19 (0.01–0.37)	**0.04**	0.19	−0.02 (−0.20–0.16)	0.86	−0.02
Unhealthy dietary behaviours (mean days/week)	−0.07 (−0.21–0.07)	0.31	−0.11	0.00 (−0.14–0.15)	0.96	0.01
**PA and dietary behaviours at school**						
Sedentary (% per day at school)	−1.51 (−2.96–−0.06)	0.05	−0.20	−1.22 (−0.26–0.19)	0.10	−0.17
Light PA (% per day at school)	0.70 (−0.51–1.92)	0.29	0.11	0.54 (−0.67–1.74)	0.39	0.09
MVPA (% per day at school)	0.76 (0.29–1.24)	**<0.01**	0.34	0.67 (0.22–1.12)	**<0.01**	0.29
Minimal two food types during lunch ** (% yes)	2.98 (1.59–5.61)	**<0.01**	na	0.62 (0.36–1.05)	0.08	na
School water consumption (0–3)	1.17 (0.95–1.38)	**<0.01**	1.14	−0.02 (−0.23–0.20)	0.86	−0.02
**PA and dietary behaviours at home**						
Sedentary (% per day at home)	−0.47 (−1.92–0.99)	0.53	−0.06	1.33 (−0.07–2.72)	0.06	0.18
Light PA (% per day at home)	0.33 (−0.74–1.41)	0.55	0.06	−1.24 (−2.27–−0.21)	**0.02**	−0.23
MVPA (% per day at home)	0.16 (−0.50–0.82)	0.63	0.05	−0.08 (−0.71–0.54)	0.79	−0.02
Minimal two food types during breakfast ** (% yes)	0.95 (0.58–1.57)	0.85	na	1.16 (0.71–1.89)	0.54	na

* For convenience purposes, this table also includes the results from the previously conducted study on overall effects of HPSF on children’s PA and dietary behaviours [17]. The number of children was not always the same in the analyses due to differences in the methods of and response to the measurements. The number of children in each statistical test is reported in the previously conducted study. ** Binary outcome, B-value is presented as odds ratio. Significance level: *p* < 0.05. Abbreviations: HPSF: Healthy Primary School of the Future; CI: confidence interval; *p*: *p*-value; ES: effect size; PA: physical activity; MVPA: moderate-to-vigorous physical activity; na: not applicable.

**Table 2 nutrients-11-02119-t002:** Effect modifiers of HPSF on overall intervention effects.

	Full HPSF vs. Control			Partial HPSF vs. Control		
	B (95% C.I.)	*p*	ES	B (95% C.I.)	*p*	ES
**Gender**						
Sedentary (% per day)	0.27 (−1.89– 2.43)	0.81		−0.01 (−2.15–2.12)	0.99	
Light PA (% per day)	−1.03 (-2.73–0.68)	0.24		0.21 (−1.49–1.90)	0.81	
MVPA (% per day)	0.77 (−0.13–1.67)	**0.09**		−0.20 (−1.09–0.69)	0.66	
Boys	0.81 (0.13–1.49)	**0.02**	0.34	0.09 (−0.57–0.74)	0.80	0.04
Girls	0.04 (-0.57–0.65)	0.89	0.02	0.29 (−0.32–0.90)	0.36	0.12
Healthy dietary behaviours (mean days/week)	−0.16 (−0.52–0.20)	0.37		0.29 (−0.07–0.65)	0.11	
Unhealthy dietary behaviours (mean days/week)	0.16 (−0.12–0.45)	0.26		0.18 (−0.11–0.46)	0.23	
**Study year**						
Sedentary (% per day)	0.02 (−2.22–2.26)	0.98		2.18 (−0.07–4.42)	**0.06**	
Study years 1–4	−1.25 (−2.72–0.21)	0.09	−0.21	0.81 (−0.59–2.20)	0.26	0.13
Study years 5–8	−1.28 (−3.00–0.45)	0.15	−0.21	−1.37 (−3.14–0.40)	0.14	−0.23
Light PA (% per day)	−0.04 (−1.77–1.70)	0.97		−1.61 (−2.53–0.69)	0.08	
Study years 1–4	0.85 (−0.33–2.02)	0.16	0.18	−0.73 (−1.63–0.17)	0.20	−0.16
Study years 5–8	0.88 (−0.48–2.25)	0.21	0.19	0.88 (0.15–1.61)	0.23	0.19
MVPA (% per day)	0.01 (−0.92–0.93)	0.99		−0.56 (−1.51–0.38)	0.24	
Healthy dietary behaviours (mean days/week)	−0.11 (−0.49–0.27)	0.58		0.01 (−0.38–0.39)	0.97	
Unhealthy dietary behaviours (mean days/week)	−0.04 (−0.34–0.26)	0.81		0.12 (−0.18–0.43)	0.43	
**Weight status**						
Sedentary (% per day)	0.93 (−2.05–3.92)	0.54		0.82 (−1.86–3.49)	0.55	
Light PA (% per day)	−0.74 (−3.06–1.58)	0.53		−0.94 (−3.05–1.17)	0.38	
MVPA (% per day)	−0.19 (−1.45–1.07)	0.76		0.10 (−1.04–1.24)	0.86	
Healthy dietary behaviours (mean days/week)	−0.18 (−0.72–0.36)	0.50		−0.16 (−0.63–0.30)	0.49	
Unhealthy dietary behaviours (mean days/week)	−0.13 (−0.55–0.28)	0.53		0.07 (−0.30–0.43)	0.73	
**SES ***						
Sedentary (% per day)		≥0.52			≥0.40	
Light PA (% per day)		≥0.57			≥0.87	
MVPA (% per day)		≥0.67			**≥0.05**	
Lowest tertile	0.29 (−0.61–1.19)	0.52	0.12	−0.51 (−1.36–0.33)	0.24	−0.21
Middle tertile	0.25 (−0.62–1.11)	0.57	0.10	0.72 (−0.11–1.55)	0.09	0.30
Highest tertile	0.50 (−0.26–1.25)	0.20	0.21	−0.21 (−0.56–0.97)	0.60	0.09
Healthy dietary behaviours (mean days/week)		≥0.26			≥0.43	
Unhealthy dietary behaviours (mean days/week)		≥0.13			**≥0.01**	
Lowest tertile	0.05 (−0.23–0.32)	0.73	0.07	0.04 (−0.21–0.30)	0.75	0.06
Middle tertile	*0.05 (*−*0.21–0.30)*	*0.73*	*0.07*	*0.23 (*−*0**.02–0.48)*	*0.07*	*0.35*
Highest tertile	−*0.22 (*−*0.44–0.01)*	*0.06*	−*0.33*	−*0.24 (*−*0**.48–0.00)*	*0.05*	−*0.36*
**PA-related parental practices ***						
Sedentary (% per day)		≥0.26			≥0.62	
Light PA (% per day)		≥0.41			≥0.53	
MVPA (% per day)		≥0.30			≥0.20	
**Nutrition-related parental practices ***						
Healthy dietary behaviours (mean days/week)		≥0.45			≥0.66	
Unhealthy dietary behaviours (mean days/week)		≥0.69			≥0.10	

* The subgroups of SES and parenting practices are more than two (SES: low/middle/high; Parenting practices: four different clusters). To investigate the potential moderation, pairwise comparisons were conducted. The lowest *p*-value of the interaction terms was presented in the table (≥*p*-value). A complete overview of all interaction terms of each pairwise comparison is presented in Appendix A. Analyses were conducted by linear mixed model analyses (continuous outcomes) or generalised estimating equations (binary outcomes), adjusted for baseline, gender, study year at T0, SES, ethnicity, and BMI z-score at T0. Significance level for the interaction term: *p* < 0.10. Significance level for each specific category of the subgroups with a significant interaction term (highlighted in green): *p* < 0.05. Abbreviations: HPSF: Healthy Primary School of the Future; CI: confidence interval; p: p-value; ES: effect size; PA: physical activity; MVPA: moderate-to-vigorous physical activity.

**Table 3 nutrients-11-02119-t003:** Effect modifiers of HPSF on intervention effects at school.

	Full HPSF vs. Control			Partial HPSF vs. Control		
	B (95% C.I.)	p	ES	B (95% C.I.)	p	ES
**Gender**						
Sedentary (% per day at school)	−0.06 (−1.52–1.40)	0.97		−1.33 (−4.03–1.38)	0.35	
Light PA (% per day at school)	0.15 (−2.07–2.36)	0.90		1.36 (−1.01–3.73)	0.26	
MVPA (% per day at school)	−0.06 (−0.94–0.81)	0.89		−0.03 (−0.94–0.84)	0.94	
Minimal two food types during lunch ** (% yes)	1.15 (.32–4.10)	0.83		1.04 (0.35–3.07)	0.94	
School water consumption (0–3)	−0.14 (−0.57–0.28)	0.51		−0.63 (−1.06–−0.20)	**<0.01**	
Boys	1.24 (0.94–1.53)	**<0.01**	1.21	0.26 (−0.03–0.55)	0.08	0.26
Girls	1.09 (0.78–1.40)	**<0.01**	1.07	−0.37 (−0.68–−0.05)	**<0.01**	−0.36
**Study year**						
Sedentary (% per day at school)	3.11 (1.63–4.59)	**0.04**		2.58 (1.10–4.05)	**0.08**	
Study years 1–4	−0.07 (−1.91–1.77)	0.94	−0.01	−0.01 (−0.92–0.90)	0.99	0.00
Study years 5–8	−3.18 (−5.30–−1.06)	**<0.01**	−0.40	−2.59 (−3.78–−1.40)	**0.03**	−0.32
Light PA (% per day at school)	−2.47 (−0.373–−1.20)	**0.05**		−2.75 (−4.01–−1.50)	**0.03**	
Study years 1–4	−0.53 (−2.09–1.02)	0.52	−0.08	−0.61 (−1.67–0.46)	0.44	−0.09
Study years 5–8	1.93 (0.56–3.30)	0.06	0.29	2.15 (1.13–3.16)	**0.03**	0.32
MVPA (% per day at school)	−0.68 (−1.59–0.23)	0.15		0.16 (−0.70–1.03)	0.73	
Minimal two food types during lunch ** (% yes)	0.90 (0.24–3.35)	0.87		0.99 (0.33–2.98)	0.99	
School water consumption (0–3)	0.32 (−0.15–0.79)	0.19		−0.51 (−0.99–−0.03)	**0.04**	
Study years 1–4	1.31 (0.94–1.69)	**<0.01**	1.28	−0.38 (−0.76–0.00)	0.05	−0.37
Study years 5–8	0.99 (0.71–1.27)	**<0.01**	0.97	0.13 (0−0.16–0.42)	0.38	0.12
**Weight status**						
Sedentary (% per day at school)	−1.62 (−5.73–2.50)	0.44		0.74 (−2.90–4.37)	0.58	
Light PA (% per day at school)	0.98 (−2.48–4.45)	0.58		−1.16 (−4.23–192)	0.46	
MVPA (% per day at school)	0.66 (−0.65–1.96)	0.32		0.44 (−0.70–1.59)	0.45	
Minimal two food types during lunch ** (% yes)	0.68 (0.12–3.99)	0.67		2.33 (0.52–10.36)	0.27	
School water consumption (0–3)	−0.14 (−0.35–0.07)	0.19		−0.02 (−0.23–0.20)	0.89	
**SES ***						
Sedentary (% per day at school)		≥0.22			≥0.48	
Light PA (% per day at school)		≥0.42			≥0.54	
MVPA (% per day at school)		≥0.12			≥0.13	
Minimal two food types during lunch ** (% yes)		≥0.22			≥0.60	
School water consumption (0–3)		≥0.26			≥0.15	
**PA-related parental practices ***						
Sedentary (% per day at school)		≥0.83			≥0.37	
Light PA (% per day at school)		≥0.88			≥0.42	
MVPA (% per day at school)		≥0.58			≥0.21	
**Nutrition-related parental practices ***						
Minimal two food types during lunch ** (% yes)		≥0.24			≥0.32	
School water consumption (0–3)		≥0.37			≥0.47	

* The subgroups of SES and parenting practices are more than two (SES: low/middle/high; Parenting practices: four different clusters). To investigate the potential moderation, pairwise comparisons were conducted. The lowest p-value of the interaction terms was presented in the table (≥*p*-value). A complete overview of all interaction terms of each pairwise comparison is presented in Appendix A. ** Binary outcome: A generalised estimating equation is used. Interaction term is Exp(B), which is the odds ratio of the first subgroup (e.g., boys) divided by the odds ratio of the second subgroup (e.g., girls), in which the odds ratio of the second group (girls) is the reference group. Analysed by linear mixed model analyses (continuous outcomes) or generalised estimating equations (binary outcomes), adjusted for baseline, gender, study year at T0, SES, ethnicity, and BMI z-score at T0. Significance level for the interaction term: *p* < 0.10. Significance level for each specific category of the subgroups with a significant interaction term (highlighted in green): *p* < 0.05. Abbreviations: HPSF: Healthy Primary School of the Future; CI: confidence interval; *p*: *p*-value; ES: effect size; PA: physical activity; MVPA: moderate-to-vigorous physical activity; na: not applicable.

**Table 4 nutrients-11-02119-t004:** Effect modifiers of HPSF on intervention effects at home.

	Full HPSF vs. Control			Partial HPSF vs. Control		
	B (95% C.I.)	*p*	ES	B (95% C.I.)	*p*	ES
**Gender**						
Sedentary (% per day at home)	1.22 (−1.59–4.04)	0.39		0.55 (−2.22–3.31)	0.70	
Light PA (% per day at home)	−1.32(−3.40–0.76)	0.21		−0.20 (−2.24–1.85)	0.85	
MVPA (% per day at home)	0.12 (−1.14–1.39)	0.85		−0.33 (−1.58–0.91)	0.60	
Minimal two food types during breakfast ** (% yes)	0.41 (.22–0.77)	0.15		0.38 (0.14–1.04)	0.12	
**Study year**						
Sedentary (% per day at home)	−2.22 (−5.16–0.73)	0.14		0.88 (−2.08–3.84)	0.56	
Light PA (% per day at home)	−1.17 (−1.03–3.36)	0.30		−0.45 (−2.65–1.76)	0.69	
MVPA (% per day at home)	1.07 (−0.24–2.38)	0.11		−0.43 (−1.74–0.88)	0.52	
Minimal two food types during breakfast ** (% yes)	1.12 (0.31–4.04)	0.86		1.23 (0.62–2.43)	0.76	
**Weight status**						
Sedentary (% per day at home)	2.27 (−2.01–6.55)	0.30		3.03 (−0.55–6.61)	**0.10**	
Non-overweight	0.00 (−1.60–1.61)	1.00	0.00	1.90 (0.33–3.47)	**0.02**	0.26
Overweight	−2.27 (−6.18–1.64)	0.26	−0.31	−1.13 (−4.32–2.06)	0.49	−0.15
Light PA (% per day at home)	−1.46 (−4.58–1.67)	0.36		−2.65 (−5.29–−0.01)	**0.05**	
Non-overweight	0.01 (−1.17–1.20)	0.99	0.00	−1.74 (−2.90–−0.58)	**<0.01**	−0.33
Overweight	1.47 (−1.40–4.32)	0.31	0.28	0.91 (−1.44–3.26)	0.45	0.17
MVPA (% per day at home)	−0.88 (−2.75–1.00)	0.36		−0.38 (−1.98–1.23)	0.65	
Minimal two food types during breakfast ** (% yes)	1.03 (0.20–5.21)	0.97		3.98 (0.87–18.33)	**0.08**	
Non-overweight	1.16 (0.59–2.29)	0.32	na	0.71 (0.35–1.40)	0.32	na
Overweight	1.20 (0.28–5.15)	0.81	na	2.81 (0.72–10.92)	0.14	na
**SES ***						
Sedentary (% per day at home)		**≥0.05**			**≥0.02**	
Lowest tertile	0.96 (−1.59–3.51)	0.46	0.13	3.36 (0.93–5.79)	**<0.01**	0.46
Middle tertile	0.26 (−2.26–2.77)	0.84	0.04	1.47 (−0.96–3.90)	0.24	0.20
Highest tertile	−2.44 (−4.80–−0.09)	**0.04**	−0.34	−0.50 (−2.85–1.85)	0.68	−0.07
Light PA (% per day at home)		**≥0.03**			**≥0.03**	
Lowest tertile	−1.18 (−3.06–0.70)	0.22	−0.23	−2.70 (−4.50–0.91)	**<0.01**	−0.52
Middle tertile	0.43 (−1.42–2.29)	0.65	0.08	−1.21 (−3.00– 0.59)	0.19	−0.23
Highest tertile	1.70 (−.04–3.44)	0.06	0.32	0.11 (−1.63–1.84)	0.91	0.02
MVPA (% per day at home)		**≥0.06**			≥0.15	
Lowest tertile	0.20 (−0.95–1.35)	0.74	0.06	−0.68 (−1.78–0.42)	0.22	−0.20
Middle tertile	−0.66 (−1.80–0.47)	0.25	−0.19	−0.26 (−1.35–0.83)	0.64	−0.08
Highest tertile	0.79 (−.27–1.85)	0.14	0.23	0.42 (−0.63–1.48)	0.43	0.12
Minimal two food types during breakfast ** (% yes)		≥0.12			≥0.12	
**PA-related parental practices ***						
Sedentary (% per day at home)		≥0.30			≥0.77	
Light PA (% per day at home)		≥0.34			≥0.47	
MVPA (% per day at home)		≥0.21			≥0.46	
**Nutrition-related parental practices ***						
Minimal two food types during breakfast ** (% yes)		≥0.46			≥0.37	

* The subgroups of SES and parenting practices are more than two (SES: low/middle/high; Parenting practices: four different clusters). To investigate the potential moderation, pairwise comparisons were conducted. The lowest *p*-value of the interaction terms was presented in the table (≥p-value). A complete overview of all interaction terms of each pairwise comparison is presented in Appendix A. ** Binary outcome: A generalised estimating equation is used. Interaction term is Exp(B), which is the odds ratio of the first subgroup (e.g., boys) divided by the odds ratio of the second subgroup (e.g., girls), in which the odds ratio of the second group (girls) is the reference group. Analysed by linear mixed model analyses (continuous outcomes) or generalised estimating equations (binary outcomes), adjusted for baseline, gender, study year at T0, SES, ethnicity, and BMI z-score at T0. Significance level for the interaction term: *p* < 0.10. Significance level for each specific category of the subgroups with a significant interaction term (highlighted in green): *p* < 0.05. Abbreviations: HPSF: Healthy Primary School of the Future; CI: confidence interval; p: p-value; ES: effect size; PA: physical activity; MVPA: moderate-to-vigorous physical activity; na: not applicable.

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
