# Peer review of "Unravelling the Effects of the Healthy Primary School of the Future: For Whom and Where Is It Effective?"

_nutrients, 2019, doi:10.3390/nu11092119_

Round 1
Reviewer 1 Report
This study examined the effects of HPSF on children’s dietary and PA behaviors at school and at home while investigating whether child characteristics or the home context moderated these effects. Overall, there are some implications for future strategies aiming to promote children’s PA and dietary behaviors. Having said this, I do have some questions and remarks about the study and the manuscript.
Line 50-58: this paragraph flows poorly and seems like you need a few sentences to transition into the next paragraph.
Line 59-64: the description of HPSF can be moved to study design.
Line 64-68: I am getting confused with these sentences. The authors mentioned they found a favorable effect on both healthy nutrition and PA, but also indicated that the two intervention schools that focused only on PA found no effects.
Line 112: Please provide a diagram to interpret the sample size for each participating school.
Line 154: where did the monitor put? Waist or wrist? In addition, as mentioned, included children are 4-12 years old, but Evenson’s cut-points apply to 5 to 8 years of age. Please provide the rationale of using such threshold. Last, it is unclear how did the data were processed (e.g., wear time define, use mins or percent of the time in each intensity?)
Line 160 and 172: do these two subtitle need to be the same?
Line 190: please clarify random and fix effects.
Line 212: a descriptive table is needed. If possible at all, please also provide bar charts for multiple comparisons.
Line 223-224: were you saying overall, at school, or at home trend?
Line 247: there was no table 2a, there were 3 tables 2
Line 329-330: don’t know why the authors mentioned this.
Line 333-335: as full and partial HPSF focused on different things, how come the authors can conclude that full HPSF had a more equal beneficial effect?
Line 344: this statement is arbitrary.
Line 3345: what did you mean by different needs? Did you mean PA patterns vary by age? What is quantitative evidence saying? Are there any other reasons can explain this? Maybe children in different grades have a different class schedule?
Line 346-348: why the authors mentioned “they are appropriate for all children” based on the finding that older children benefited more?
Line 356-360: it is not surprised as children from low SES families tend to have less access to PA equipment and safe PA environment.
Any implications?
Author Response
Dear editor,
Thank you for allowing us to submit a revised version of our manuscript. We also want to thank the reviewers for their helpful comments to further improve the manuscript. Below we explain how we incorporated the comments in our revision. In the manuscript, we have marked the changes in yellow.
Reviewer 1
Comments and Suggestions for Authors
This study examined the effects of HPSF on children’s dietary and PA behaviors at school and at home while investigating whether child characteristics or the home context moderated these effects. Overall, there are some implications for future strategies aiming to promote children’s PA and dietary behaviors. Having said this, I do have some questions and remarks about the study and the manuscript.
Line 50-58: this paragraph flows poorly and seems like you need a few sentences to transition into the next paragraph.
We agree that the transition into the next paragraph could be improved. We have changed it as follows:
Page 2, Line 59-61: The ‘Healthy Primary School of the Future’ (HPSF) is a Dutch initiative that aims to improve the health and well-being of all children in the school by sustainably integrating health and well-being within the whole school system (14, 15).
Line 59-64: the description of HPSF can be moved to study design.
We understand the reviewer’s suggestion. However, the reason for the short description of HPSF in the Introduction section is to provide the reader with a bit of information about HPSF before describing the overall effects of HPSF.
Line 64-68: I am getting confused with these sentences. The authors mentioned they found a favorable effect on both healthy nutrition and PA, but also indicated that the two intervention schools that focused only on PA found no effects.
This is indeed the case. The two intervention schools that focused on both healthy nutrition and PA showed several favourable effects. The two schools that focused solely on PA showed no significant effects on healthy behaviours. One of the main explanations for these results is that simultaneously addressing nutrition and PA creates a synergistic effect leading to larger effects. Both the findings and possible explanations are discussed in detail in the study of Bartelink et al.1. We have adjusted the sentence to avoid confusion as follows:
Page 2, Line 65-69: The overall study includes, among others, an extensive process evaluation and several effect evaluations. One of the effect evaluations found favourable intervention effects on children’s health behaviours in the two intervention schools that focused on both healthy nutrition and PA (17). The two intervention schools that focused only on PA found no effects, also not on children’s PA behaviours.
1 Bartelink NH, van Assema P, Kremers SP, Savelberg HH, Oosterhoff M, Willeboordse M, et al. One-and Two-Year Effects of the Healthy Primary School of the Future on Children’s Dietary and Physical Activity Behaviours: A Quasi-Experimental Study. Nutrients. 2019;11(3):689.
Line 112: Please provide a diagram to interpret the sample size for each participating school.
We have included the sample sizes per school of the number of children who are included in the analyses in Table S1 in Additional file 2.
Line 154: where did the monitor put? Waist or wrist? In addition, as mentioned, included children are 4-12 years old, but Evenson’s cut-points apply to 5 to 8 years of age. Please provide the rationale of using such threshold. Last, it is unclear how did the data were processed (e.g., wear time define, use mins or percent of the time in each intensity?)
We agree with the reviewer about the limitations of the Evenson’s cut-off points. However, by our knowledge no cut-off points exist for the broad age range we have included in our study. We preferred to use the same cut-off points for all children, also at follow-up, to ensure that the changes in PA are not due to different cut-off points. Therefore, we decided to use the Evenson’s cut-off, even though it has its limitations. Furthermore, we agree with the reviewer that some information is missing about this measurement. We have added this information to the manuscript, see Page 4, Line 158-168. We used the percentage per day for each activity level, which is included in the tables.
Page 4, Line 158-168: Children’s PA levels – accelerometry: At the beginning of the measurement week all participating children from study years two to eight received an accelerometer for seven days (Actigraph GT3X+, 30Hz, 10s epoch). The monitor was attached to the hip with an elastic band and had to be worn all day except while sleeping or during activities in which water was involved (e.g., swimming, bathing and showering). The accelerometry data were processed using ActiLife version 6.13.3. Wear time validation was assessed using Choi’s classification criteria (25). Minimal wear time was defined as 480 min per day between 6 a.m. and 11 p.m. (26). The first day of measurement was excluded to prevent reactivity (27). Measurements containing at least three weekdays (after excluding the first measurement day) and one weekend day were used in the analyses (28). The activity levels, classified using Evenson’s cut-off points, were in counts-per-minute (CPM) (29): sedentary behaviour (SB; ≤100 CPM), light PA (LPA; 101 – 2295 CPM), and moderate-to vigorous PA (MVPA; ≥2296 CPM).
Line 160 and 172: do these two subtitle need to be the same?
We decided to make the first part of the subtitle the same to show that children’s dietary behaviours have been measured by different instruments: a parent-reported questionnaire and two child-reported questionnaires.
Line 190: please clarify random and fix effects.
We agree with the reviewer that the information about the models is limited. We added the following to the manuscript:
Page 5, Line 199-206: Linear mixed-model analyses (continuous outcomes) and generalized estimating equations (binary outcomes) were used (see Bartelink et al. (17)) for the overall outcomes and school- and home-specific outcomes. Since measurements were repeated within participants, we used a two-level model with measurements as the first level and participants as the second level. The fixed part of the model consisted of group (full HPSF, partial HPSF, control), time (T0, T1, T2) and the interaction terms of group with time. We were not able to include class as a level in the model, because commonly more than one division of a class existed, e.g., 4a or 4b, and children often did not have fixed class divisions for all years.
Line 212: a descriptive table is needed. If possible at all, please also provide bar charts for multiple comparisons.
Since the article already includes several tables, we have added the descriptives table to the additional file (Additional file 2). This table is derived from the other evaluation study in which we investigated the effects of HPSF on children’s BMI z-score. The sample sizes are identical between the two studies.
Line 223-224: were you saying overall, at school, or at home trend?
The trends were found at school. We have added this information to the manuscript.
Page 5, Line 238-239: Favourable trends were found for both sedentary time (full HPSF: ES=-.20; partial HPSF ES=-.17) and light PA at school (full HPSF: ES=.11; partial HPSF ES=.09).
Line 247: there was no table 2a, there were 3 tables 2
We would like to thank the reviewer for noticing this. We have changed it into Table 2, Table 3, and Table 4.
Line 329-330: don’t know why the authors mentioned this.
We agree with the reviewer that this is not a result that should be mentioned in the Discussion section. Therefore, we have deleted the sentence.
Line 333-335: as full and partial HPSF focused on different things, how come the authors can conclude that full HPSF had a more equal beneficial effect?
This conclusion was based on our finding that the percentage of significant moderators was lower in the full HPSF compared to the partial HPSF. This indicates that the health-promoting changes as part of the full HPSF succeeded more in creating a favorable effect in all children, not just a subgroup of them.
Line 344: this statement is arbitrary.
We do not intent to make a statement, but we want to provide a possible explanation. We changed the sentence as follows:
Page 13, Line 368-369: This may indicate that the activities in school were more appropriate for the older children.
Line 345: what did you mean by different needs? Did you mean PA patterns vary by age? What is quantitative evidence saying? Are there any other reasons can explain this? Maybe children in different grades have a different class schedule?
We understand the reviewer’s confusion. We did not mean by ‘different needs’, the need to move more or less, but the need for different types of PA activities or for PA-promoting aspects. Previous research has for example shown that older children’s activity levels were more negatively affected by the number of peers present, while younger children were more negatively affected by the number of supervisors.
To prevent the confusion in the manuscript, we added this example to the text and made it more explicit. We have done this as follows:
Page 13-14, Line 367-375: The effects on children’s PA behaviours at school were moderated by study year in both the full and partial HPSF: older children benefitted consistently more from HPSF than younger children. This may indicate that the activities in school were more appropriate for the older children. This is in line with the results and conclusions of several studies that indicated that children of different ages have different needs regarding PA-activities (6, 34). Previous research has for example shown that older children’s activity levels were more negatively affected by the number of peers present, while younger children were more negatively affected by the number of supervisors (35). Therefore, it is recommended when implementing PA-related activities to ensure that either they are appropriate for all children or that age-specific PA-activities are implemented.
Line 346-348: why the authors mentioned “they are appropriate for all children” based on the finding that older children benefited more?
We would like to refer to the previous comment. We meant to say that a closer look should be given to the content of the PA-related activities to ensure that activities are used that either are appropriate for all ages or that age-specific activities are used (see Page 13-14, Line 367-375).
Line 356-360: it is not surprised as children from low SES families tend to have less access to PA equipment and safe PA environment.
Any implications?
We totally agree with the reviewer. These results show that a systems approach is needed to improve the PA behaviours of children from these low SES families. This means that not only children’s PA behaviours in school should be improved, but also in the other microsystems of a child, such as home and neighborhood. This implicates that school health promotion efforts should include these microsystems in their health-promoting changes, e.g., by homework assignments that include parents and/or focus directly on health promotion in the home context. In the Discussion we refer to these microsystems, see Page 14, Line 399-409
Reviewer 2 Report
Minor spell check required: Authors may want to consider using the standard American English spellings of words such as neighborhood, behavior, favorable.
Author Response
Dear editor,
Thank you for allowing us to submit a revised version of our manuscript. We also want to thank the reviewers for their helpful comments to further improve the manuscript. Below we explain how we incorporated the comments in our revision. In the manuscript, we have marked the changes in yellow.
Reviewer 2
Minor spell check required: Authors may want to consider using the standard American English spellings of words such as neighborhood, behavior, favorable.
In accordance to our other publications in Nutrients we used the UK English for spellings of words in this manuscript.
Reviewer 3 Report
Thank you for the opportunity to review this manuscript which presents exploration of intervention effects by location (at school/at home) and of potential modifying factors.
Major issues:
1. There is a lack of detail on how the control schools were selected, whether the response rate differed between schools, and how the data collection periods were determined. How were the control and intervention schools monitored for any relevant changes over the study period?
2. L397: Not having randomised selection is still a problem, although it can’t be corrected. Motivation of schools might be expected (of course) to be an effect modifier, and all the correlates of motivation of schools are likely to be distributed unequally between groups. Because all the effect measures are presented in relation to students of the control schools, one wonders what part of the measured effects were a result of the program(s) and what were related to other (probably school based) factors.
3. Multiple statistical comparisons mean that some of the observed statistical differences are likely to have arisen by chance – this should be acknowledged in the strengths and weaknesses.
4. The measurement of breakfast intake and lunch intake appears to have been undertaken for some children, not all the children included in the analyses (analyses included children who were in years 1 to 7 at baseline, meals were measured for children in years 3 to 8 and 4 to 8 respectively). Does this mean in the number of children in the analyses was different for these items? Could this be acknowledged/included in the relevant tables?
5. The dietary aspects of the full HPSF intervention (described as free healthy lunch, water policy and water bottle) seems likely to be the direct reason for the difference in effects on dietary behaviours seen (compared to the control schools and different to the partial HPSF i.e. improved lunch, improved water consumption, slightly improved healthy dietary behaviour). Because these might be considered direct consequential results of the intervention, it is perhaps not a surprise that there are few or no effect modifiers for these relationships (all students get a free lunch, the school water policy applies to everyone). The dietary measurements were quite simple (due in part to the age, and large number of subjects), and the authors have mentioned that the limitation of using questionnaires. The authors are clearly interested in the impact of the intervention to the home environment (or the influence of the home environment on the intervention). Therefore, as an extension, future research might investigate dietary intake more comprehensively and using a method less reliant on recollection (for example an image collection method), at least for a sub-group of the study group.
Minor issues:
L35: A primary source reference should be used to support the opening statement that dietary and PA habits are formed at a young age.
L67: presumably the issue is that the earlier analysis only presented overall effects
L80: ‘higher in prevalence’ rather than ‘highly prevalent’.
L124 – I imagine the data collection and processing were identical (rather than similar) to the earlier paper – otherwise it would be important to outline what was different.
L126 – why was ethnicity considered important enough to adjust for but not considered a potential effect modifier?
L134 – the cited reference doesn’t appear to be directly relevant to the SES score – the earlier study (ref no 17) cites a different reference. What are the reasons for this? How were single parent households treated?
L215 – because the number of children included was 1676, greater than the number of children joining at baseline (1403), the explanation makes better sense if worded ‘only including children who were eligible for study years one to seven at baseline …’
Table 1 -double asterisk is missing from the dichotomous variables (i.e. where OR are given).
At present there are three Table 2s (starting on pages 7, 9 and 11 respectively). From the text, these are probably intended to be Tables 2a, 2b and 2c. My preference would be to name them Table 2, 3 and 4 for clarity. All tables could include the information that the models were adjusted for gender, study year at baseline, ethnicity, SES, and childrens BMI z-score at baseline unless these were the subject of the specific analysis (as understood from the statistical analyses section).
Multiple statistical comparisons mean that some of the observed statistical differences are likely to have arisen by chance – this should be acknowledged in the strengths and weaknesses.
The measurement of breakfast intake and lunch intake appears to have been undertaken for some children, not all the children included in the analyses (analyses included children who were in years 1 to 7 at baseline, meals were measured for children in years 3 to 8 and 4 to 8 respectively). Does this mean in the number of children in the analyses was different for these items? Could this be acknowledged/included in the relevant tables?
The dietary aspects of the full HPSF intervention (described as free healthy lunch, water policy and water bottle) seems likely to be the direct reason for the difference in effects on dietary behaviours seen (compared to the control schools and different to the partial HPSF i.e. improved lunch, improved water consumption, slightly improved healthy dietary behaviour). Because these might be considered direct consequential results of the intervention, it is perhaps not a surprise that there are few or no effect modifiers for these relationships (all students get a free lunch, the school water policy applies to everyone). The dietary measurements were quite simple (due in part to the age, and large number of subjects), and the authors have mentioned the limitation of using questionnaires. The authors are clearly interested in the impact of the intervention to the home environment (or the influence of the home environment on the intervention). Therefore, as an extension, future research might investigate dietary intake more comprehensively and using a method less reliant on recollection (for example an image collection method), at least for a sub-group of the study group.
Author Response
Dear editor,
Thank you for allowing us to submit a revised version of our manuscript. We also want to thank the reviewers for their helpful comments to further improve the manuscript. Below we explain how we incorporated the comments in our revision. In the manuscript, we have marked the changes in yellow.
Reviewer 3
Thank you for the opportunity to review this manuscript which presents exploration of intervention effects by location (at school/at home) and of potential modifying factors.
Major issues:
There is a lack of detail on how the control schools were selected, whether the response rate differed between schools, and how the data collection periods were determined. How were the control and intervention schools monitored for any relevant changes over the study period?We agree with the reviewer that some information is missing. We included the information about the response rate at baseline in Table 1S, in Additional file 2. More detailed information about the control schools, the data collection, and monitoring is described in several other publications on the HPSF studies. To provide the reader with more information or a direct reference, we have added the following to the manuscript.
Page 2, Line 63-68: HPSF is being investigated in an overall study among four intervention and four control schools by a multi-disciplinary research group (14, 15). The overall study includes, among others, an extensive process evaluation and several effect evaluations. One of the effect evaluations found favourable intervention effects on children’s health behaviours in the two intervention schools that focused on both healthy nutrition and PA (17).
Page 4, Line 86-88: Ethical approval (14-N-142) for the overall study was given by the Medical Ethics Committee Zuyderland located in Heerlen (Parkstad, the Netherlands). A detailed description of the overall study and the recruitment of the eight schools is reported in Willeboordse et al (15).
Page 3, Line 125-126: Data were gathered annually during one week of measurements, conducted in September-November of 2015 (T0, previous to the start of HPSF in November 2015), 2016 (T1) and 2017 (T2).
Page 5, Line 229-232: Due to the selection used for the current study, i.e., children were eligible for the current research only when they were in study years one to seven at baseline and did not switch schools, 1676 children were included in the analyses, see Additional file 2 for characteristics of the study sample.
L397: Not having randomised selection is still a problem, although it can’t be corrected. Motivation of schools might be expected (of course) to be an effect modifier, and all the correlates of motivation of schools are likely to be distributed unequally between groups. Because all the effect measures are presented in relation to students of the control schools, one wonders what part of the measured effects were a result of the program(s) and what were related to other (probably school based) factors.
The reviewer describes an important aspect here. Schools are all different from each other; they all have their own context. Motivation is an important aspect that can influence the effects, but also school climate, practices of teachers, or dominating organisational issues. The moderating role of the school context on the effects of HPSF is the main focus of another study 2. In the current study, we aimed to analyze the data on the level of the child and investigate for whom and where HPSF is effective.
2 Bartelink, N., van Assema, P., Jansen, M., Savelberg, H., & Kremers, S. (2019). The Moderating Role of the School Context on the Effects of the Healthy Primary School of the Future. International journal of environmental research and public health, 16(13), 2432.
Multiple statistical comparisons mean that some of the observed statistical differences are likely to have arisen by chance – this should be acknowledged in the strengths and weaknesses.
We have changed the manuscript as follows:
Page 14-15, Line 419-426: The quasi-experimental design can be seen as a limitation of the study, since we were unable to (cluster-) randomize schools. To deal with this limitation, we controlled for BMI z-score at T0, gender, study year at T0, SES score, and ethnicity in all analyses. However, despite the lack of randomization, the design enabled us to test the effectiveness in terms of differences in children’s health behaviours between the three school groups over time, and we were able to enrol schools on the basis of motivation, which reflects the real-life situation of school health promotion. Another limitation is the multiple statistical testing in this study, which may increase the likelihood that the observed statistical differences have arisen by chance.
The measurement of breakfast intake and lunch intake appears to have been undertaken for some children, not all the children included in the analyses (analyses included children who were in years 1 to 7 at baseline, meals were measured for children in years 3 to 8 and 4 to 8 respectively). Does this mean in the number of children in the analyses was different for these items? Could this be acknowledged/included in the relevant tables?
This is indeed the case. Since we have included these numbers in a previous study and do not want to extend the tables even further, we have decided to acknowledge this aspect and refer to the publication on the previously conducted study.
Page 7, Line 253-256: For convenience purposes, this table also includes the results from the previously conducted study on overall effects of HPSF on children’s PA and dietary behaviours (17). The number of children was not always the same in the analyses due to differences in the methods of and response to the measurements. The number of children in each statistical test are reported in the previously conducted study.
The dietary aspects of the full HPSF intervention (described as free healthy lunch, water policy and water bottle) seems likely to be the direct reason for the difference in effects on dietary behaviours seen (compared to the control schools and different to the partial HPSF i.e. improved lunch, improved water consumption, slightly improved healthy dietary behaviour). Because these might be considered direct consequential results of the intervention, it is perhaps not a surprise that there are few or no effect modifiers for these relationships (all students get a free lunch, the school water policy applies to everyone). The dietary measurements were quite simple (due in part to the age, and large number of subjects), and the authors have mentioned that the limitation of using questionnaires. The authors are clearly interested in the impact of the intervention to the home environment (or the influence of the home environment on the intervention). Therefore, as an extension, future research might investigate dietary intake more comprehensively and using a method less reliant on recollection (for example an image collection method), at least for a sub-group of the study group.
We totally agree with the reviewer and have added this in the manuscript as follows.
Page 15, Line 438-441: In general, the assessment of children’s dietary behaviours had its limitations as only questionnaires were used, which are subjective measurements and may lead to socially desirable answers (41). To assess children’s dietary behaviours more objectively, future research could include image collection methods.
Minor issues:
L35: A primary source reference should be used to support the opening statement that dietary and PA habits are formed at a young age.
L134 – the cited reference doesn’t appear to be directly relevant to the SES score – the earlier study (ref no 17) cites a different reference. What are the reasons for this? How were single parent households treated?
We would like to thank the reviewer for noticing. Some errors existed in the references. We have checked the whole manuscript and have corrected the errors.
L67: presumably the issue is that the earlier analysis only presented overall effects L80: ‘higher in prevalence’ rather than ‘highly prevalent’. L124 – I imagine the data collection and processing were identical (rather than similar) to the earlier paper – otherwise it would be important to outline what was different. L215 – because the number of children included was 1676, greater than the number of children joining at baseline (1403), the explanation makes better sense if worded ‘only including children who were eligible for study years one to seven at baseline …’ Table 1 -double asterisk is missing from the dichotomous variables (i.e. where OR are given). At present there are three Table 2s (starting on pages 7, 9 and 11 respectively). From the text, these are probably intended to be Tables 2a, 2b and 2c. My preference would be to name them Table 2, 3 and 4 for clarity. All tables could include the information that the models were adjusted for gender, study year at baseline, ethnicity, SES, and children’s BMI z-score at baseline unless these were the subject of the specific analysis (as understood from the statistical analyses section).
We would like to thank the reviewer for his/her detailed look at the manuscript and that he/she identified these issues.
L126 – why was ethnicity considered important enough to adjust for but not considered a potential effect modifier?
Ethnicity may influence the effects and is therefore included as a covariate in the model. However, the subgroup with children who have a non-Western ethnicity is very small, which makes it statistically difficult to analyse the separate effects for this subgroup. This also means that the subgroup with children who have a Western ethnicity is almost similar to the total group, which means that the effect for this subgroup will also be very much similar to the overall effect.
Round 2
Reviewer 1 Report
I am grateful to the authors for taking time and effort necessary to reply to my inputs over my concerns about this paper. The authors responding appropriately to my comments and my concerns have been allayed after carefully reading the revised version